# PLA_2_ Inhibitor Varespladib as an Alternative to the Antivenom Treatment for Bites from Nikolsky’s Viper *Vipera berus nikolskii*

**DOI:** 10.3390/toxins12060356

**Published:** 2020-05-29

**Authors:** Oleksandr Zinenko, Igor Tovstukha, Yevgen Korniyenko

**Affiliations:** 1V. N. Karazin Kharkiv University, 61058 Kharkiv, Ukraine; korgenia@gmail.com; 2Ukrainian Independent Ecology Institute, 61001 Kharkiv, Ukraine; igorandreevich8@gmail.com

**Keywords:** snakebite, Viperidae, therapy, PLA_2_, inhibitor, Varespladib, mouse

## Abstract

Although envenoming by a small East European species of viper is rarely severe, and only exceptionally fatal, lack of specific antivenom stocks in a few areas within this region and possible severe side effects of antivenom application leave most bites to be treated only with antihistamines and supportive therapy. Varespladib is an effective inhibitor of snake phospholipase, and, as such, it could be considered as first-line therapy. The Nikolsky’s viper venom contains an extremely high concentration of phospholipase A2 (PLA_2_), responsible for the toxic effects of the venom, as well as minor amounts of other toxins. If Varespladib can successfully inhibit PLA_2_ activity, the Nikolsky’s viper could be one of the first venomous snakes having an antitoxin-specific treatment regimen. To assess that, Varespladib was administered alone subcutaneously to adult male CD-1 mice (8 mg/kg) and compared to mice exposed to *Vipera berus nikolskii* crude venom (8 mg/kg = 10 LD_50_) or a combination of Varespladib and the same amount of the venom. Experimental animals were monitored for the presence of envenoming symptoms and mortality for 48 h after injection. Eighty percent of mice receiving both Varespladib and venom survived, while 100% of the control group receiving venom alone died within 4 h. Experimental results are consistent with Varespladib acting as an effective antitoxin in the mouse model against Nikolsky’s viper venom. Further studies are needed under experimental conditions that more closely resemble natural envenoming (i.e., delayed administration).

## 1. Introduction

Snake bites are a global health problem, mostly in tropical countries. It is estimated that about 2.7 million people are envenomed annually, and hundreds of thousands die each year [1]. The World Health Organization listed snake bites as a category A neglected tropical disease, underlining the global significance of this problem. 

A specific antivenom has become standard as an efficient antidote to the systemic effects of snakebite envenoming [2]. Disadvantages of antivenom therapy are the high or limited specificity of monovalent and polyvalent antivenoms, high production costs, the need for cold-chain delivery and storage, the requirement of professional supervised application, and a high risk for severe side effects [1]. These disadvantages hamper its use as a first-line treatment in remote settings or in areas with supply chain disruptions. Moreover, there is a growing problem in many regions, especially in Africa, where antivenom supply specific for local snake species has decreased due to lack of commercial interest [1,3].

Advances in molecular biology technologies make it possible to search for specific inhibitors of key enzymes responsible for pathogenesis from envenoming rather than using antibodies produced by donor animals [4,5,6,7,8,9]. Suppression of snake toxin activity with low molecular mass inhibitors is not a new idea, but using new methods like molecular docking [10], and facilitated by a large number of available drugs with known properties to inhibit certain enzyme activity, may substitute or aid traditional antivenom treatments in the near future.

Nikolsky’s viper (*Vipera berus nikolskii* Vedmederja, Grubant et Rudaeva, 1986) is a subspecies of the widespread common adder *Vipera berus* Linnaeus, 1758. The Nikolsky’s viper inhabits southern European broad-leafed forests across Eastern Romania, Moldavia, Ukraine, and the Southern Russian Federation [11,12]. It is locally protected in Ukraine and the Russian Federation as rare or endangered [13,14], but local populations nevertheless are often dense, and the snake is often found in considerable numbers in rural areas, in gardens, near summer houses, and in parks [13]. Currently, antivenom is not produced in Ukraine; however, antivenom to *Vipera berus berus* is produced in the Russian Federation and used in all cases of envenoming, including those which happen in southern regions of Eastern Europe where *V. b. nikolskii* lives. The effectiveness of this type of treatment has not been estimated, but similarities in venom composition between these taxa predict that common adder antivenom may inhibit at least some of the fractions of the venom of Nikolsky’s viper [15]. Statistics for the number of bites annually is not available; however, media reports suggest they are not rare. Its bite is not usually lethal, and illness resolves after several days of symptomatic treatment in the hospital, but the burden on the public health system’s resources can be greatly alleviated if venom-specific therapeutics such as Varespladib are available for treatment.

The venom of Nikolsky’s viper is well studied [15,16,17,18,19]. The most abundant enzymes are phospholipases A2 (PLA_2_) (65% of dry mass), followed by serine protease (19% of dry mass) [15], making it one of the most PLA_2_-rich venom among venomous snakes [20]. Our observations indicated that the main symptoms of envenoming by *V. b. nikolskii* consisted of local edema and pain, lymphangitis, and hypotension. However, local necrosis, blistering, or hematoma were not observed. Mild neurotoxic activity was demonstrated in in vivo experiments with HDP-2 PLA_2_ from the venom of *V. b. nikolskii* [18]. The crude venom also had an effect on cranial nerves and caused progressive limb paralysis resulting in flaccidity in mice [19]. The murine LD50 of crude venom is 0.80 mg/kg [19], similar to the observed murine LD50 in the sister subspecies *V. b. berus* (i.e., 0.86 mg/kg) [19,21]. *V. b. nikolskii* has only one prevalent peptide in the venom, PLA_2_ [15], making it an ideal subject to examine new approaches in snakebite treatment using specific inhibitors [4,5]. The venom of the common adder *V. b. berus* has several components and demonstrates edema-inducing, hemorrhagic, and neuro-, myo-, cyto-, hemotoxic and enzymatic activities, and it has slightly less PLA_2_ (59%) compared to Nikolsky’s viper [21,22]. 

Varespladib LY315920 was found to be a specific inhibitor of vertebrate PLA_2_ [5] and may fit most requirements for first-line treatment of many venomous species bites. It suppresses in vitro activity of PLA_2_ from multiple species of venomous snakes from different systematic groups and protects mice (both increases survival and postpones onset of symptoms) from Viperid (*Vipera berus*, *Deinagkistrodon acutus*, *Agkistrodon halys*) and Elapid (*Micrurus fulvius* and *Oxyuranus scutellatus*) venom in experimental conditions [5,6,7,8]. Recent studies also have shown that Varespladib can neutralize anticoagulant activity of African spitting cobras that are not neutralized by conventional antivenom [9].

In this study, we expanded on the experiment by Lewin et al. [5] by testing the venom from *Vipera berus nikolskii* to determine whether Varespladib treatment would protect mice and mitigate symptoms of envenoming by *V. b. nikolskii* venom. 

## 2. Results

All three mice in the positive control group (venom only) died within four hours (26–290 min) after injection (Figure 1). The onset of symptoms, which included first disturbance, rapidly progressed (3–30 min) into hypodynamia, loss of reaction to external stimuli, depression and alteration of movements, weakness, loss of coordination, difficult breathing, tachypnoea with involvement from auxiliary breathing muscles, cyanotic skin on snout and feet, and venous hyperemic skin on ears. Necropsy demonstrated repletion of all internal organs with blood, hyperemic mesenteric veins, pulmonary edema and hyperemia, signs of blood circulation disorders in kidneys and liver, and signs of heart failure (see Appendix A).

Mice in the treatment group (venom + Varespladib), similarly to the positive control group, demonstrated suppressed activity and loss of normal reaction to external stimuli after injection, but to a lesser extent. However, three of five mice recovered to normal levels within 24 h after injection. Similar to the positive control group, symptoms developed gradually in one mouse 12 h after the injection and resulted in death 24 h after injection (Figure 1). Necropsy of dead animals from the treatment group demonstrated repletion of internal organs with blood (less pronounced than in the first group), pathology of the liver, and point hemorrhages in kidney parenchyma (Appendix A). The cause of death was heart failure with signs of disseminate blood coagulation syndrome. One more mouse in the treatment group developed ataxia and paresis of the hind limbs and altered breathing with auxiliary muscle involvement within 13 h after the injection; however, all symptoms resolved gradually in 48 h.

A statistically significant difference was seen in survival analyses when positive control and treatment groups were compared using Gehan’s Wilcoxon (*p* = 0.02), Cox’s *F* test (*p* = 0.0045), and Cox–Mantel (*p* = 0.004) tests. Median life expectancy in the positive control group exposed to venom was 52 ± 30 min post injection. Median life expectancy cannot be calculated properly in the treatment group because 80% of the mice survived beyond the observation period for the experiment. Mice in the control group did not show any signs of pathology, and 100% of the animals survived through the duration of the experiment.

## 3. Discussion

This is the first time we tested the protective action of Varespladib against *V. b. nikolskii* venom. This taxon has one of the highest levels of PLA_2_ in the venom [20]; therefore, single-component antidote therapy could be a practical procedure for the treatment of *V. b. nikolskii* bites. Our study proves that this concept works on laboratory animals and may be used as a prospective treatment to replace antivenom therapy, alleviating the shortage of this biological product for the treatment of venomous bites in this taxon.

The dose of venom administered in our experiment was almost ten times the LD50 for *V. b. berus* or *V. b. nikolskii,* and this explains the high mortality in the control group [19,21]. Similar to the experiment of Lewin et al. [5] with *V. b. berus* venom, Varespladib was an effective treatment to envenoming with *V. b. nikolskii*. However, under the same conditions (dose, route of administration, regime) the survival rate was higher, and onset of symptoms started later than in the experiment with *V. b. berus* venom. This could be due to the fact that more toxins, like serine proteases or L-amino-acid oxidase, exist in the venom from the common adder *V. b. berus* that are not inhibited by Varespladib [21]. The venom of *V. b. nikolskii* has 65% of PLA_2_ in dry mass [15] compared to the 60% of all phospholipases present in *V. berus* venom [22]; thus, better efficiency of Varespladib, which inhibits specifically this enzyme in this case, was expected. The administration of Varespladib may have also contributed to the differences observed. Lewin et al. [5] injected Varespladib in a solution of bicarbonate/dextrose sodium, while we used a 1% dimethyl sulfoxide (DMSO) solution, which produced an opalescent suspension of Varespladib. Interaction between DMSO and Varespladib could be complex and could affect the pharmacological dynamics of Varespladib. The interaction between DMSO and Varespladib, or direct pharmacological actions of DMSO, could explain the mixture’s resulting prolonged action, longer protection, lower mortality, and later onset of symptoms of envenoming when compared to the experimental results found by Lewin et al. [5].

PLA_2_ in snake venom and Varespladib may have different deactivation times. Varespladib’s half-life in humans is five hours when infused intravenously [23] and could be up to 12 times less in mice [24]. The interaction of Varespladib with venom PLA_2_ can potentially delay the drug’s deactivation and extend its half-life in animals; however, significant improvement of the envenomed animals’ conditions after repetitive administration of Varespladib [6] may favor a more rapid deactivation time of Varespladib than PLA_2_. Different routes of administration, solubilization conditions, and solvent used together may also alter Varespladib activity time. In the experiment by Lewin et al. [5], Varespladib could have been quickly metabolized or excreted and led to the appearance of symptoms in most of the experimental mice, while in our study, activity of Varespladib in mice could have been prolonged, or drug absorption increased, due to usage of DMSO, thus making it possible to postpone signs of envenoming caused by PLA_2_ activity. This longer intoxication may explain the observed liver and kidney pathologies in the mice from the treatment group that died. This hypothetical mechanism of interaction is supported by additional experiments by Lewin et al. [6,7], when repetitive administration of Varespladib improved protection of experimental animals against venoms of Elapid snakes and reversed symptoms of envenoming.

While promising as a future therapy of envenoming by Nikolsky’s viper in humans, further testing and adjustment of the route of administration and chemical form of Varespladib must be done. As tested, Varespladib LY315920 is highly soluble in DMSO, but not in water, and subcutaneous or intramuscular injection of large volumes is not advisable in humans, especially taking into account the possible coagulopathic conditions following snake envenoming. Alternatively, Varespladib methyl (LY333013) was tested in clinical trials too [23] and has a number of advantages compared to Varespladib (LY315920). These advantages include high water solubility, which makes possible parenteral administration in field conditions or intravenous administration in a hospital setting. Adjustment of dose and the scheme of treatment under realistic conditions of envenoming in humans are essential future studies for the clinical application of this proposed therapy.

Varespladib is an effective treatment in mice against the *V. b. nikolskii* venom, and no visible pathologies have been observed in the negative control group. Since this drug has been shown to be safe, or did not shown adverse effects when administered intravenously in humans, even during a continuous period up to a few weeks (see overview in [23]), it may serve as a potential treatment for snakebites and a substitute for complement antivenom treatments of snakebites in Ukraine. However, rigorous clinical trials to demonstrate its clinical efficiency are still necessary. Snakebite treatment with Varespladib does not require continuous administration, unlike other pathologies listed in the overview [23], and likely will not be provided to patients with such copathologies. Therefore, it can be presumed that that Varespladib will likely be safe under the same conditions. Bites from *V. b. nikolskii* could become the first to be treated for Viperid snakes using a novel, inhibitor-based approach, together with *Elapidae* snakes tested in similar experiments [6,7]. In cases of envenoming by other local snakes (e.g., *V. b. berus* and *V. renardi*), where the concentration of PLA_2_ is lower than in *V. b. nikolskii* but still remains a main component of the venom with the proportion of dry mass of venom between 44% in *V. renardi* [15] and 60% in *V. berus* [22], Varespladib may still be an efficient treatment capable of protecting from PLA_2_-associated signs of envenoming. In the case of more complex venoms, other enzymes or toxin inhibitors, such as proteinase inhibitors or other antitoxins, are likely to be used simultaneously to prevent local cytotoxic and hemorrhagic effects [1,4].

## 4. Conclusions

In conclusion, the high PLA_2_ levels in the venom and the results discussed herein strongly support that Varespladib demonstrates potential to be utilized as an effective drug against envenoming by the widespread Nikolsky’s viper *Vipera berus nikolskii*. Further assessment of Varespladib in human studies has the potential to confirm the utility of this drug as a therapeutic of choice against Nikolsky’s viper venom and other snake species. The use of this drug could also help relieve the need for antivenom production and storage in Ukraine.

## 5. Materials and Methods

### 5.1. Animals and Reagents

Male CD-1 mice, 3–5 weeks old, were used in the experiments with body weights ranging between 18–22 g. Venom was obtained from wild *V. nikolskii* from the Kharkiv region of Ukraine and dried over desiccant at room temperature. Prior to administration, the dry venom was stored in cold, dry conditions at 10 °C and resuspended in distilled water just before the experiment. Animal experiments were performed in agreement with the U.K. Animals (Scientific Procedures) Act, 1986 and the European Communities Council Directive of 24 November 1986 (86/609/EEC), and they received an approval from the Bioethics Committee of Kharkiv National University (Protocol #7, 21.09.2017).

### 5.2. Treatment Conditions

CD-1 mice were randomly divided into three groups: (a) a positive control group consisting of three mice injected subcutaneously (s.c.) with 8 mg/kg of *V. b. nikolskii* venom in distilled water; (b) a treatment group of five mice receiving 8 mg/kg of *V. b. nikolskii* venom in water (equivalent to 10 lethal dose 50 (LD_50_)) [19] and, immediately after venom injection, dosed separately with another injection of 8 mg/kg Varespladib LY315920 in 1% DMSO (Sigma-Aldrich, St. Louis, MO, USA) subcutaneously; and (c) a negative control group of five mice receiving s.c. 8 mg/kg Varespladib alone. The methodology utilized repeated a previously published experiment with *V. berus* venom [5] in order to obtain compatible results and verify the effect of Varespladib on a new snake taxon. All groups were observed for 48 h after treatment. Animals were monitored for symptoms of disease, responses to stimuli, and time of death was logged for each animal. Animals succumbing to intoxication were kept in the refrigerator until necropsy on the same day.

### 5.3. Data Analysis

A survival analysis was performed comparing groups receiving venom alone and groups receiving venom and Varespladib treatment. Groups were compared using non-parametric Gehan’s Wilcoxon, Cox’s *F* test, and Cox–Mantel tests using Statistica 6.0.

## Figures and Tables

**Figure 1 toxins-12-00356-f001:**
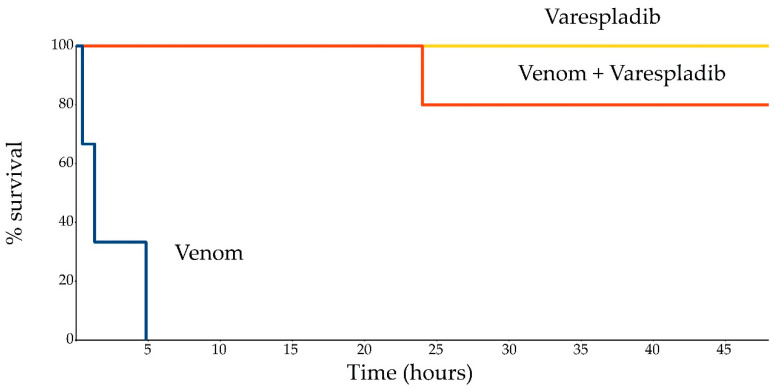
In vivo protection with Varespladib LY315920 in CD-1 mice envenomed with *V. b. nikolskii* venom. The treatment groups were divided into venom alone (blue line), treatment alone (yellow line), and venom exposure plus treatment with Varespladib (orange line).

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
