# Peer review of "PLA2 Inhibitor Varespladib as an Alternative to the Antivenom Treatment for Bites from Nikolsky’s Viper Vipera berus nikolskii"

_toxins, 2020, doi:10.3390/toxins12060356_

Round 1

Reviewer 1 Report

Very nice addition to the literature. I have no major concerns about this revision. Some minor points: 

Too many significant figure: p=0.02, p=0045 etc. should be sufficient. 53+/-30 minutes is sufficient. 

A statically significant difference was seen in survival analyses when positive control and 110 treatment groups were compared using Gehan’s Wilcoxon (p=0.02128), Cox's F test (p=0.00453) and 111 Cox-Mantel (p=0.00425) tests. Median life expectancy in the positive control group exposed to 112 venom was 52.73±30.44 min post injection. Median life expectancy cannot be calculated properly in 113 the treatment group because 80% of the mice survived beyond the observation period for the 114 experiment. Mice in the control group did not show any signs of pathology and 100% of the animals 115 survived through the duration of the experiment.

139 different deactivation times. Varespladib's half-life 

Comment

Even if the 1/2 life of the drug free in serum could be short, its affinity for venom could be quite a bit stronger and have a greater effective 1/2 life. 

Author Response

Thank you for the review. Here is our response:

"Very nice addition to the literature. I have no major concerns about this revision. Some minor points: 

Too many significant figure: p=0.02, p=0045 etc. should be sufficient. 53+/-30 minutes is sufficient. 

A statically significant difference was seen in survival analyses when positive control and 110 treatment groups were compared using Gehan’s Wilcoxon (p=0.02128), Cox's F test (p=0.00453) and 111 Cox-Mantel (p=0.00425) tests. Median life expectancy in the positive control group exposed to 112 venom was 52.73±30.44 min post injection. Median life expectancy cannot be calculated properly in 113 the treatment group because 80% of the mice survived beyond the observation period for the 114 experiment. Mice in the control group did not show any signs of pathology and 100% of the animals 115 survived through the duration of the experiment."

- Have reduced number digits in significance level and time indication

"139 different deactivation times. Varespladib's half-life 

Comment

Even if the 1/2 life of the drug free in serum could be short, its affinity for venom could be quite a bit stronger and have a greater effective 1/2 life."

- Thank you, agree. Have added sentence to discuss this (Lines 192-197), however any facts which supports or rejects this are currently unavailable.

Reviewer 2 Report

The authors provided the revision of the manuscript and now it is ready to be accepted.

Author Response

"The authors provided the revision of the manuscript and now it is ready to be accepted."

- Many thanks for your input.

Reviewer 3 Report

I have no more comments.

Author Response

"I have no more comments."

- Thank you very much for your input and review.

This manuscript is a resubmission of an earlier submission. The following is a list of the peer review reports and author responses from that submission.

Round 1

Reviewer 1 Report

This is a nicely executed study extending what is known about varespladib. I have a few major comments and a few minor comments (typos), but all should be easily addressed and should not hold up publication should the article be accepted by the editorial board. My comments are longer than what is necessary to make changes--merely included for discussion. 

Major Comments:

It may be useful for the authors to discuss how they envision the clinical use of the drug (oral/IV) as opposed to SQ, which is not a likely final clinical application outside of veterinary use. Subcutaneous administration is very convenient for mice, but even with the very good solubility of the drug’s water soluble sodium salt (20mg/ml) the volume would likely be too high for human use as, for example, with an IM, SQ or autoinjector type presentation and there would be risk of injury in a coagulopathic patient. The SQ approach may be very useful for veterinary application (e.g. working military canines) where it could be injected in large volume under the loose skin of the neck. For clinical use/future studies could include dose finding remains to be confirmed—for rapid development, confirming that the dosed used in clinical trials is approximately the same would be highly useful. It would be interesting to test the oral prodrug (LY333013) which is readily dissolved in 8% gum Arabic/water in this snake (Lewin et al 2018).

There are several references I would regard as more appropriate to suggest the safety of this class of drugs. VISTA-16 (Cited by the authors) was a 16-week trial and the drug was given orally in patients with acute coronary syndrome + kidney disease or diabetes. It’s a bit apples and oranges in terms of comparison. They terminated the trial because of efficacy issues (www.clinicaltrials.gov), but they did not conclude the drug was safe, either. The major highlight in terms of differences is that for snakebite, administration would be Acute and Short term—not 16 weeks and most patients will not have significant copathology (e.g. recent angioplasty for acute coronary syndrome + diabetes or kidney disease).  

I think the closest article in humans is this one with parenteral (IV) administration. The drug has not been administered SQ in humans although this is the manner we did this in our original publication as reported by the authors.

Abraham E, Naum C, Bandi V, et al. Efficacy and safety of LY315920Na/S-5920, a selective inhibitor of 14-kDa group IIA secretory phospholipase A2, in patients with suspected sepsis and organ failure. Crit Care Med 2003; 31: 718–28.

This review: Adis R&D Profile. Varespladib. Am J Cardiovasc Drugs 2011; 11: 137–43. This article very efficiently details 25 or so clinical trials with varespladib. I have attached the PDF because is not available without a fee. I believe elements of several trials, including VISTA-16 were performed in Ukraine (I am not sure).

Other articles I think are significantly worth citing:

Recent findings by other groups:

Mátyás A. Bittenbinder et al. Toxins (Basel). 2018 Dec; 10(12): 516. 10.3390/toxins10120516  Coagulotoxic Cobras: Clinical Implications of Strong Anticoagulant Actions of African Spitting Naja Venoms That Are Not Neutralised by Antivenom but Are by LY315920 (Varespladib)

Yiding Wang, Jing Zhang, Denghong Zhang, Huixiang Xiao, Shengwei Xiong, Chunhong Huang. Exploration of the Inhibitory Potential of Varespladib for Snakebite Envenomation Molecules. 2018 Feb; 23(2): 391.

Neurotoxicity of V. berus group

One potentially useful discussion point is neurotoxicity, especially in pediatric cases. Are cases in children more likely to be severe?

Malina T, Babocsay G, Krecsák L, Erdész C. Further clinical evidence for the

existence of neurotoxicity in a population of the European adder (Vipera berus berus) in eastern Hungary: second authenticated case. Wilderness Environ Med. 2013 Dec;24(4):378-83. doi: 10.1016/j.wem.2013.06.005..

Malina T, Krecsák L, Jelić D, Maretić T, Tóth T, Siško M, Pandak N. First

clinical experiences about the neurotoxic envenomings inflicted by lowland

populations of the Balkan adder, Vipera berus bosniensis. Neurotoxicology. 2011 Jan;32(1):68-74.

Minor Comments:

Current WHO terminology for snakebite is “envenoming”—equivalent to “poisoning” rather than “poisonation”. It’s probably time to unify the literature, though this is not a major point.

82: Venon (typo)

126: Serine protease (typo)

159: likely is probably more appropriate word than “need”. There is not much literature to support this comment (or refute it, but the only article that actually looks at this suggests otherwise, Wang et al, citation below).

176 CD-1 (typo)

DMSO, delivery: Again, might be worth commenting that the form that will likely make it to the clinic is the sodium salt, which is very soluble in water. DMSO is the appropriate solvent for varespladib HCl—highly insoluble in water.

Reviewer 2 Report

PLA2 inhibitor Varespladib as an alternative to the antivenom treatment for bites from Nikolsky’s viper Vipera berus nikolskii.

The Authors presented an interesting approach to inhibit the phospholipase A2 (PLA2) effects from the Vipera berus Nikolskii venom using Varespladib drug.

However the major concern of this investigation is the poor quality of results that have been presented. The reader would have really appreciated to see some “data not shown” results, to really trust the survival effect of the drug.

Really few animals were used, at least discussed in this paper, to generate a statistically significant difference on the presented data. Why the author used only 3 animals for the group a) and 5 for both b) and c) groups? 

There are no indication about the gender and the age.

There are no indications how the combination Venom+Varespladib was prepared, i.e: for how long the two compounds were incubated together before injection? Have the author considered to do a Venom:Varespladib titration?

Minor comment. Revise the text for several typing error mistakes, for example: Venon vs Venom.

Line 110, doesn’t fit with the text.

Reviewer 3 Report

The present study was to study the protective effect of Varespladib against the toxicity of Vipera berus nikolskii crude venom in mice.  The authors found that Varespladib can effectively reduce the lethality of Vipera berus nikolskii crude venom in mice.  Although the result is impressive, several points should be revised further.

Some in vitro experiments should be conducted for confirming the inhibitory effect of Varespladib on PLA2 enzymes in Vipera berus nikolskii crude venom.

To mimic the real condition, the mice should be challenged with Vipera berus nikolskii crude venom for 1-4 h before administration of Varespladib.  Then the protective effect of of Varespladib against the toxicity of Vipera berus nikolskii crude venom in mice is assessed.